# Fabrication of Electrochemical Influenza Virus (H1N1) Biosensor Composed of Multifunctional DNA Four-Way Junction and Molybdenum Disulfide Hybrid Material

**DOI:** 10.3390/ma14020343

**Published:** 2021-01-12

**Authors:** Jeong Ah Park, Jinmyeong Kim, Soo Min Kim, Hiesang Sohn, Chulhwan Park, Tae-Hyung Kim, Jin-Ho Lee, Min-Ho Lee, Taek Lee

**Affiliations:** 1Department of Chemical Engineering, Kwangwoon University, 20, Kwangwoon-Ro, Seoul 01897, Korea; m3m0719@gmail.com (J.A.P.); wls629@icloud.com (J.K.); k-soomin@hotmail.com (S.M.K.); hsohn@kw.ac.kr (H.S.); chpark@kw.ac.kr (C.P.); 2School of Integrative Engineering, Chung-Ang University, 84, Heukseok-ro, Seoul 06974, Korea; thkim0512@cau.ac.kr; 3School of Biomedical Convergence Engineering, Pusan National University, 49, Busandaehak-ro, Yangsan 50612, Korea; leejh@pusan.ac.kr

**Keywords:** electrochemical biosensor, DNA 4WJ, influenza virus, H1N1, molybdenum disulfide

## Abstract

The outbreak of the influenza virus (H1N1) has symptoms such as coughing, fever, and a sore throat, and due to its high contagious power, it is fatal to humans. To detect H1N1 precisely, the present study proposed an electrochemical biosensor composed of a multifunctional DNA four-way junction (4WJ) and carboxyl molybdenum disulfide (carboxyl-MoS_2_) hybrid material. The DNA 4WJ was constructed to have the hemagglutinin aptamer on the head group (recognition part); each of the two arms has four silver ions (signal amplification part), and the tail group has an amine group (anchor). This fabricated multifunctional DNA 4WJ can specifically and selectively bind to hemagglutinin. Moreover, the carboxyl-MoS_2_ provides an increase in the sensitivity of this biosensor. Carboxyl-MoS_2_ was immobilized using a linker on the electrode, followed by the immobilization of the multifunctional 4WJ on the electrode. The synthesis of carboxyl-MoS_2_ was confirmed by field emission scanning electron microscopy (FE-SEM), and the surface of the electrode was confirmed by atomic force microscopy. When H1N1 was placed in the immobilized electrode, the presence of H1N1 was confirmed by electrochemical analysis (cyclic voltammetry, electrochemical impedance spectroscopy). Through selectivity tests, it was also possible to determine whether this sensor responds specifically and selectively to H1N1. We confirmed that the biosensor showed a linear response to H1N1, and that H1N1 could be detected from 100 nM to 10 pM. Finally, clinical tests, in which hemagglutinin was diluted with human serum, showed a similar tendency to those diluted with water. This study showed that the multi-functional DNA 4WJ and carboxyl-MoS_2_ hybrid material can be applied to a electrochemical H1N1 biosensor.

## 1. Introduction

The influenza virus, commonly known as the flu, is a highly acute respiratory disease, and includes the 1918 Spanish influenza, as well as the 2009 H1N1 influenza, affecting numerous patients and causing deaths worldwide [1]. The World Health Organization (WHO) reported 250,000–500,000 deaths annually worldwide due to seasonal influenza viruses [2]. The high infectivity of the H1N1 virus is due to its epidemiological nature. In most cases, the spread of H1N1 is due to indirect contact transmission or droplet transmission of the virus in an infected patient’s coughing and sneezing directly into the lower respiratory tract of another person [3,4]. Therefore, in urban areas, where there are many direct and indirect contacts between people, disease spread can occur more quickly and cause considerable human illness. In addition, antiviral drugs should be administered within 48 h of the occurrence of clinical symptoms, as infants, the elderly, and patients with chronic diseases may experience aggravation of underlying disease or pneumonia [5,6,7]. Therefore, a fast and accurate H1N1 confirmatory test is necessary to address these issues.

Current H1N1 diagnostic methods in hospitals and healthcare centers include reverse transcription polymerase chain reaction (RT-PCR), cell culture, and rapid antigen tests. In all three methods, a sample is taken from the nasopharynx using a cotton swab and diagnosed [8]. RT-PCR and cell culture are highly sensitive and accurate but require a long time to confirm and involve expensive equipment [9,10]. The rapid antigen test is inexpensive and can provide a diagnosis within 30 min but is less sensitive and accurate and cannot be used for confirmation [11,12,13,14]. Therefore, to make a H1N1 biosensor practical in the field, fast signal response time, accuracy, and high sensitivity are required. To satisfy these conditions, we intend to diagnose H1N1 virus by electrochemical (EC) detection. This method detects and analyzes physical and chemical signals generated by the interaction of the analyte to be detected with bioreceptors through a signal converter [15,16,17]. It is easy to carry and reacts quickly with the analyte to be measured, so it has the advantage of quick results and is suitable for diagnosing rapidly spreading viruses.

The aim of this study was to propose a multifunctional bioprobe-based biosensor composed of a multifunctional DNA four-way junction (4WJ) and carboxyl-molybdenum disulfide (MoS_2_) for diagnosis of the H1N1 virus by the EC detection method. The multifunctional DNA 4WJ is composed of four single-stranded DNA (ssDNA) that perform various functions [18,19]. The H1N1 aptamer and amine group were added to this 4WJ in the head and tail parts. The DNA aptamer specifically binds to a target molecule [20,21]. Advantages include easy modification to the required form, low cost of production using chemical synthesis, high stability, and inexpensive manufacturing [22,23,24]. By introducing an aptamer to 4WJ-a and linking an amine group to 4WJ-b, the bioprobe was able to bind to the carboxyl group of carboxyl-MoS_2_, detecting H1N1. In addition, by adding silver ions to the base pair mismatched with cytosine–cytosine (C–C), it was possible to stabilize the mismatched base pair and amplify the electrochemical signal [25,26]; the sensitivity of the sensor was increased by introducing carboxyl-MoS_2_. The synthesis of carboxyl-MoS_2_ was verified by field-emission scanning electron microscopy (FE-SEM), and the immobilization process of each step was confirmed by atomic force microscopy (AFM). The performance of the proposed sensor was demonstrated by cyclic voltammetry (CV) and electrochemical impedance spectroscopy (EIS). Figure 1 shows the overall schematic diagram of the fabricated biosensor.

## 2. Materials and Methods

### 2.1. Materials

The influenza A H1N1 (A/New Caledonia/20/1999) Hemagglutinin/HA Protein (His Tag) and influenza A H5N1 (A/VietNam/1203/2004) Hemagglutinin/HA Protein (His Tag and fragment crystallizable (Fc) Tag) were purchased from Sino Biological (Beijing, China). Cysteamine, myoglobin, hemoglobin, bovine serum albumin (BSA), potassium hexacyanoferrate(III) and potassium hexacyanoferrate(II) trihydrate ([Fe(CN)_6_]^3−/4−^), and C-reactive protein (CRP) were purchased from Sigma–Aldrich (St. Louis, MO, USA). Silver nitrate (AgNO_3_) and 4-(2-hydroxyethyl)-1-piperazineethanesulfonic acid (HEPES) were purchased from Daejung Chemical and Metals Co., Ltd. (Siheung, South Korea). For the clinical test, the human serum from human male AB plasma (USA origin) was purchased from Sigma–Aldrich (St. Louis, MO, USA). The four fragments of DNA 4WJ were synthesized by Bioneer (Daejeon, South Korea). All oligonucleotides were supplied by Bioneer and diluted in nuclease-free water. Table 1 showed the DNA sequences were as follows:

### 2.2. Synthesis of Carboxyl-MoS_2_

To synthesize carboxyl-modified MoS_2_, the following experiment was performed [27]. A total of 1.5 g of sodium molybdate dihydrate was dissolved in 50 mL of deionized water (DIW) for 30 min, with stirring at 800 rpm, at 60 °C. The temperature and rpm were maintained until the end of the experiment, and the flask inlet was closed with foil. Next, 4-ammonium benzoic acid (3.0 g) was dissolved in a 45 mL of solution prepared using ethanol (30 mL), and DIW (15 mL) was slowly added for carboxyl modification. The solution melted when left at room temperature and, when turbid, the tube was soaked in water until it cleared. After 30 min, sodium sulfide nonahydrate (3.5 g) dissolved in DIW (10 mL) was added. After another 30 min, hydrazine hydrate (2 mL) dissolved in DIW (4 mL) was added to the flask. After waiting for 30 min, one side of the flask was closed with a rubber stopper and the other side was connected to a pump. Next, 5 mL of hydrochloric acid (2 mL) dissolved in DIW (28 mL) was slowly added to the flask to confirm that the solution was black. We waited 4–6 h with the pump connected. After synthesis was completed, the flask was cooled at room temperature for approximately 1 h, and a 200-nm filter paper was placed in a glass funnel to filter the solution. The particles stuck to the filter paper were collected using DIW. The particles on the filter paper after complete filtration were dried at room temperature for a day.

### 2.3. Assembly of Multifunctional DNA 4WJ

DNA 4WJ was assembled to serve as the proposed biosensor multifunctional bioprobe and to simplify the detection step. H1N1 HA aptamer-tagged 4WJ-a was introduced to detect the HA protein (H1N1 aptamer/4WJ-a), and amine group-tagged 4WJ-b was introduced for immobilization (NH_2_/4WJ-b). A C–C mismatch was used for all 4WJ (a, b, c, d), and silver ions were inserted into the C-C mismatch for amplification of the electrochemical signal. The DNA 4WJ structure was assembled by annealing the four complementary strands of DNA (H1N1 apt/4WJ-a, NH_2_/4WJ-b, 4WJ-c, 4WJ-d) at an equal molar ratio in Tris-Magnesium-Sodium Chloride (TMS) buffer (50 mM Tris, 10 mM MgCl_2_, 100 mM NaCl) using heating for 5 min at 80 °C, followed by cooling to 4 °C at a rate of 2 °C/min. DNA 4WJ assembly was confirmed by tris-boric acid-magnesium-native polyacrylamide gel electrophoresis (8% TBM-PAGE: Tris-Boric Acid-Magnesium Polyacrylamide Gel Electrophoresis) prepared in TBM buffer (89 mM Tris, 200 mM boric acid, 2 mM MgCl_2_, pH 7.6). DNA samples were electrophoresed on each lane at 80 V for 65 min [28].

### 2.4. Fabrication of Electrode

A P-type 4-inch Si substrate (100) with a thermally oxidized SiO_2_ was initially cleaned using a piranha solution composed of H_2_SO_4_ (Daejung Chemical and Metals Co. Ltd., Siheung, South Korea) and H_2_O_2_ (Daejung Chemical and Metals Co. Ltd., Siheung, South Korea) at a volume ratio of 7:3 for 5 min at room temperature and baked on a hot plate at 200 °C for 20 min for adhesion promotion with photoresist. The pre-patterned metal deposition was prepared by a standard photolithographic process using a negative photoresist. Su-82 photoresist was spin-coated onto the substrate at 3000 rpm and then baked at 65 °C for 1 min and 95 °C for 1 min. After exposure of 110 mJ/cm2 (i-line) using an MA6 mask aligner (SUSS MicroTec, Garching Germany), the substrate was post-baked using the same steps as pre-baking and developed for 70 s. Metal deposition of Cr (2 nm) and Au (50 nm) was carried out using an electron beam evaporator. Finally, a lift-off process was performed using the remover PG at 70 °C for 2 h. The fabricated electrode was cleaned using piranha solution under the same conditions. The gap size between the working electrode and the counter electrode of the proposed biosensor was 10 μm [29]. As it is composed of 7 electrodes, it can be used 7 times, providing high fidelity of the measurement results.

For the electrochemical analysis, the working chamber was prepared using a PDMS linkage and attached to the electrode. After attachment of the chamber to the electrode, the substrates were washed with ethanol and DIW to remove organic contaminants, which could have been induced in the curing process of PDMS in a high-temperature chamber [30]. The design of the electrode was produced by referring to Lee group’s report [31].

### 2.5. Fabrication of Multifunctional DNA 4WJ/Carboxyl-MoS_2_ Heterolayer

To prepare multifunctional DNA 4WJ/carboxyl-MoS_2_, the previously prepared electrode was sonicated with 99.5% acetone for 5 min. Next, it was washed with ethanol and DIW and dried with N_2_ gas to remove the residue. To connect the carboxyl-MoS_2_ and the electrode, 3 µL of 10 mM cysteamine was reacted on the electrode for 1 h. This is because when carboxyl-MoS_2_ is added to the electrode where cysteamine is immobilized, the carboxyl group of carboxyl-MoS_2_ reacts with the amino group of cysteamine to form a peptide bond. A total of 3 µL of 0.1 mg/mL carboxyl-MoS_2_ was placed on the electrode with immobilized cysteamine and stored overnight. Subsequently, 3 µL of 1 µM DNA 4WJ, which had been subjected to annealing, was reacted on the electrode for 2 h; 3 µL of 100 nM H1N1 HA protein was then placed on the electrode and reacted for 2 h. The overall procedure was performed at room temperature. Excess biomolecules were removed by DIW and N_2_ gas stream [32].

### 2.6. Surface Morphology Analysis

Surface analysis of multifunctional DNA 4WJ on carboxyl-MoS_2_ was performed by tapping-mode AFM using Nanoscope IV/Multimode (Digital Instruments, New York, NY, USA). The bare Au substrate, carboxyl-MoS_2_-modified substrate, and multifunctional DNA 4WJ-modified substrate were investigated for comparison. The n-type doped Si phosphorous (RTESP, Bruker, Billerica, MA, USA) tip was introduced with resonance peaks. The resonance peak of the frequency response was approximately 230–305 kHz, and the spring constant of the cantilever was 20–80 N/m. In addition, the surface morphology of carboxyl-MoS_2_ immobilized on the electrode was also analyzed using FE-SEM (Auriga, Carl Zeiss, Berlin, Germany).

### 2.7. Electrochemical Analysis of Multifunctional 4WJ/Carboxyl-MoS_2_ Heterolayer on Electrode

Cyclic voltammetry (CV) and EIS experiments were performed using a three-electrode system. DNA 4WJ/carboxyl-MoS_2_/Au was used as the working electrode, and the other side of the prepared electrode was used as the counter electrode. As a reference electrode, an Ag/AgCl reference electrode (CH Instruments, Austin, TX, USA) was used. CV was performed at a scan rate of 50 mV/s and 0.6–0.1 V. EIS was measured at a frequency range of 100 kHz to 1 Hz with a DC potential of 0.25 V. A 1:1 M ratio of 5 mM [Fe(CN)_6_]^3−/4−^ redox repair in 10 mM HEPES (pH 7.04) containing 1 M KCl was used as buffer. The copper probe tips were made to contact the counter electrode and the working electrode [33]. To confirm clinical usage of the fabricated biosensor, the results of CV and EIS were compared HA (Hemmaglutinin) protein diluted in PBS (Phosphate buffered saline) buffer and HA protein diluted in 10% human serum, respectively.

## 3. Results

### 3.1. Construction of Multifunctional DNA 4WJ

In order to construct a multifunctional bioprobe that detects HA protein, it was necessary to confirm the assembly of DNA 4WJ. The 4WJ consists of four DNA strands. The H1N1 aptamer was incorporated into 4WJ-a, and the amine group was incorporated into 4WJ-b to react with the carboxyl group of carboxyl-MoS_2_. 4WJ-c,d are sequences that are complementary to the other sequences. Finally, by inserting Ag^+^ into C-C mismatches, the electrochemical signal width was expected (Figure 2a).

In this study, TBM-PAGE and TBE-PAGE (tris-borate-EDTA were performed. TBM-PAGE is a polyacrylamide gel electrophoresis using a buffer solution of Tris-HCl, boric acid, MgCl_2_ and was used to confirm that the 4WJs were well bound. TBE-PAGE is a polyacrylamide gel electrophoresis using a buffer solution of tris- HCl, boric acid, EDTA (Ethylenediaminetetraacetic acid) and was used to check that the H1N1 HA protein and aptamer were well bound. Figure 2b shows the TBM-PAGE, and Figure 2c shows the TBE-PAGE.

Figure 2b shows the PAGE results of the H1N1 HA-Apt-4WJ-a/NH_2_-4WJ-b/4WJ-c/4WJ-d assembly. The gel indicated H1N1 HA-Apt-4WJ-a (lane 2), NH_2_-4WJ-b (lane 3), 4WJ-c (lane 4), and 4WJ-d (lane 5). HA-Apt-4WJ-a/NH_2_-4WJ-b/4WJ-c/4WJ-d (lane 6) showed a change in migration, confirming the proper binding of 4WJ. In addition, Figure 2c shows HA-Apt-4WJ-a (lane 2), HA protein (lane 4), HA-Apt-4WJ-a/HA protein (lane 6), myoglobin (lane 8), and HA-Apt-4WJ-a/myoglobin (lane 10). As HA protein and myoglobin are proteins, no results could be confirmed from the DNA electrophoresis results, suggesting that the results were well analyzed. In addition, lane 6 showed a change in migration compared with lane 2, but lane 10, using myoglobin as a control group, had no change in migration compared with lane 2. It was found that the aptamer specifically and selectively bound only to the HA protein surface antigen.

### 3.2. Investigation of Fabricated HA Protein/Multifunctional DNA 4WJ on Carboxyl-MoS_2_ Heterolayer

The surface morphology of the carboxyl-MoS_2_/DNA 4WJ heterolayer immobilized on Au electrodes were analyzed using AFM. Figure 3a showed the topography of the cleaned Au electrode observed by AFM. The Ra value was 0.322 ± 0.045 nm, the RMS roughness (Rq) was 0.564 ± 0.377 nm, and the vertical distance (VD) between surfaces was 45.275 ± 9.806. Through self-assembly using cysteamine, carboxyl-MoS_2_ was immobilized on the electrode surface to observe particles larger than 1 μm (Figure 3b). That showed a similar form to carboxyl-MoS2 observed through FE-SEM. In regard to Carboxyl-MoS_2_ immobilized on electrode, Ra value was 3.392 ± 1.731, Rq value was 4.335 ± 2.776, VD value increased significantly to 469.14 ± 63.254, respectively. Compared with the multifunctional DNA 4WJ/HA protein immobilized sample (Figure 3c) and the Au substrate surface immobilized with Carboxyl-MoS_2_, Overall HA protein is covered on the surface. In addition, a spherical lump (above 40 nm) that looks bright here can be seen in the formation of the captive HA protein of DNA 4WJ. The values of Ra, Rq, and VD at this sample are 1.810 ± 0.833, 2.216 ± 0.998, and 75.814 ± 19.822, respectively. Figure 3d,e depicted the synthesized carboxyl-MoS_2_ which observed by FE-SEM. Figure 3f showed the surface roughness analysis of fabricated electrode. Surface analysis results using AFM and FE-SEM that was shown that the electrode consisting of different layers of carboxyl-MoS2 and DNA 4WJ can be applied as a biosensor for detecting HA protein.

### 3.3. Electrochemical Response of H1N1 Detection

In the immobilization step, carboxyl-MoS_2_ and 4WJ were used to increase the EC amplification and sensitivity. CV was performed at a potential ranging of 0.6–0.1 V with a scan rate of 50 mV/s in a 5 mM [Fe(CN)_6_]^3−/4−^ solution in 10 mM HEPES (pH 7.04) containing 1M KCl. Figure 4a shows the cyclic voltammogram of each immobilization step. The values of I_pc_ (cathodic peak height) and I_pa_ (anodic peak height) of the Au electrode were 5.414 ± 0.844 μA at 319.20 ± 2.815 mV and -5.203 ± 0.773 μA at 225.01 ± 9.978 mV, and I_pc_ and I_pa_ of the Au/carboxyl-MoS_2_ electrode were 5.383 ± 2.031 μA at 306.59 ± 8.877 mV and –5.659 ± 2.377 μA at 179.28 ± 4.180 mV. The values of I_pc_ and I_pa_ of the Au/carboxyl-MoS_2_/4WJ electrode were 3.620 ± 0.433 μA at 312.23 ± 7.960 mV and -3.778 ± 0.438 μA at 167.90 ± 8.351 mV, and I_pc_ and I_pa_ values of the Au/carboxyl-MoS_2_/4WJ/HA protein electrode were 3.603 ± 0.375 μA at 311.66 ± 1.480 mV and -3.760 ± 0.405 μA at 168.95 ± 6.397 mV. Figure 4b shows the EIS spectrum performed within the frequency range of 100 kHz to 1 Hz at a DC potential of 0.25 V in the same solution as that used in CV. The EIS spectrum was analyzed using a Randles equivalent circuit, where ”R_s_” stands for solution-phase resistance, ”Cdl” stands for double-layer capacitance, “R_ct_” stands for charge-transfer resistance, and “Z_w_” stands for Warburg impedance (Figure 4b and Figure 5a,c). In the order of Au, Au/carboxyl-MoS_2_, Au/carboxyl-MoS_2_/4WJ, Au/carboxyl-MoS_2_/4WJ/HA protein, the R_ct_ value is 1799.327 ± 560.25 Ω, 4021.782 ± 921.17 Ω, 6728.600 ± 1663.66 Ω, and 5928.850 ± 869.02 Ω, respectively. It could be seen that the value of R_ct_ increased in the order of Au, Au/carboxyl-MoS_2_, Au/carboxyl-MoS_2_/4WJ/HA protein, and Au/carboxyl-MoS_2_/4WJ. The tendency of these values allows the designed sensor to detect H1N1.

The EIS response was obtained according to H1N1 at various concentrations, and a 99% confidence interval was set to estimate the data. The experiment was conducted by diluting in the concentration range from 100 nM to 10 pM (Figure 5a). It could be seen that as the concentration increased, the size of R_ct_ increased. Figure 5b shows a graph of linear regression analysis using various concentrations (100 nM to 10 pM) of H1N1 on the *y*-axis and R_ct_ on the *x*-axis. In addition, the slope of this graph is 905.00 ± 75.40, and the intercept is 11,692.54 ± 787.38 (R^2^ = 0.9796). Using the related equation, the detection limit was 10 pM.

As can be seen from the CV results, the current value is increased by immobilizing carboxyl-MoS_2_ (Figure 4a. Red line). This phenomenon would be elucidated by carboxyl-MoS_2_ has a current amplification effect [27]. In addition, introducing the silver ions in 4WJ, electrochemical signal was embodied [25]. The reason why the current range is lower than carboxyl-MoS_2_ on Au substrate is due to the correlation between Au/carboxyl-MoS_2_ and 4WJ. If only 4WJ was inserted without introducing silver ions, it would have formed a smaller current range than Figure 4a. As a result, by adding carboxyl-MoS_2_ and 4WJ, the EIS tendency as shown in Figure 4b was formed, and the EIS result of Au/carboxyl-MoS_2_/4WJ/H1N1 was able to detect H1N1 up to a wider range of concentrations.

Moreover, to determine whether the designed biosensor can be used in clinical practice, EIS was measured for a concentration range of 100 nM to 10 pM using H1N1 diluted in 10% human serum (Figure 5c,d). Figure 5c shows an EIS graph for the concentration range of 100 nM to 10 pM, and it can be seen that the trend is similar to the result of the experiment (Figure 5a). The difference between Figure 5a–d is the presence or absence of serum, so the magnitude of resistance is different. Figure 5d is a graph showing R_ct_ by H1N1 concentration. The slope of this graph is 2458.98 ± 235.65, and the intercept is 32,856.03 ± 2077.60, resulting in a linear regression curve (R^2^ = 0.97, 319).

To confirm the selectivity of the biosensor, Figure 6a,b show the selectivity test designed to determine whether H1N1 (black) was selectively detected; 5 mM [Fe(CN)_6_]^3−/4−^ redox repair (1:1 M ratio) in 10 mM HEPES (pH 7.04) containing 1 M KCl was used as a buffer. The proteins used in this experiment were H5N1 (red), hemoglobin (blue), myoglobin (green), and CRP (purple) and were used as control groups. Since other proteins didn’t respond to this biosensor, it was confirmed that the designed biosensor could detect H1N1 with high selectivity and specificity.

## 4. Conclusions

H1N1 influenza virus detection is becoming increasingly important in the diagnosis and treatment of infections. In this study, the proposed EC biosensor was constructed to detect H1N1 quickly and accurately. To this end, silver ions were added to the DNA 4WJ for electrochemical amplification, and carboxyl-MoS_2_ was applied to the sensor to increase the sensitivity. In addition, the results were confirmed through a TBM-PAGE experiment to check the binding of 4WJ and a TBE-PAGE experiment to determine whether the 4WJ specifically and selectively binds only to the H1N1 surface antigen. Carboxyl-MoS_2_ was immobilized on the electrode using a linker, and 4WJ was immobilized on it. The presence of H1N1 was confirmed through electrochemical analysis (CV and EIS) when H1N1 was added to the previously immobilized electrode. In addition, it was found that detection from 100 nM to 10 pM was possible. The limit of detection was 10 pM. This can be measured in a wide range compared to other biosensors (Table 2). In addition, the synthesis of carboxyl-MoS_2_ was confirmed using SEM, and the designed particles were also confirmed by AFM. In the selectivity test, the control group was set to H5N1, hemoglobin, myoglobin, and CRP, and it was shown that this sensor reacted specifically and selectively to H1N1 only. Finally, through clinical tests, it was confirmed that the designed bioprobe can be applied in real situations. The influenza virus, generally known as the flu, spreads more rapidly in densely populated urban areas where there are many direct and indirect contacts between people and can cause a considerable amount of damage. As the biosensor designed in this study is an aptamer-based bioprobe, it is inexpensive, capable of mass-production, and widely applicable. Developing a biosensor that detects H1N1 enables simple and quick diagnosis, which will be useful in clinical experiments.

## Figures and Tables

**Figure 1 materials-14-00343-f001:**
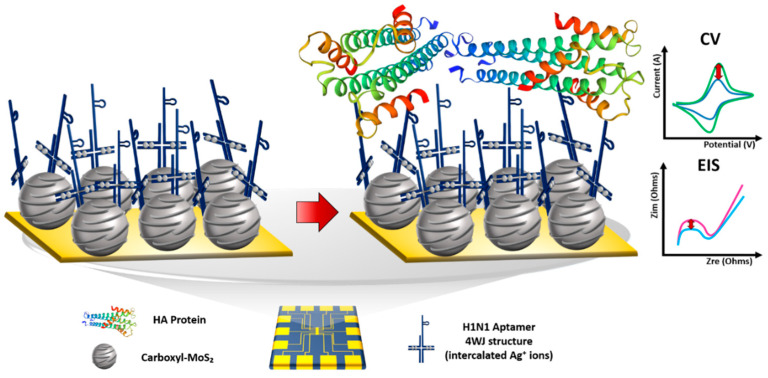
Schematic image of the fabricated electrochemical biosensor for H1N1 detection.

**Figure 2 materials-14-00343-f002:**
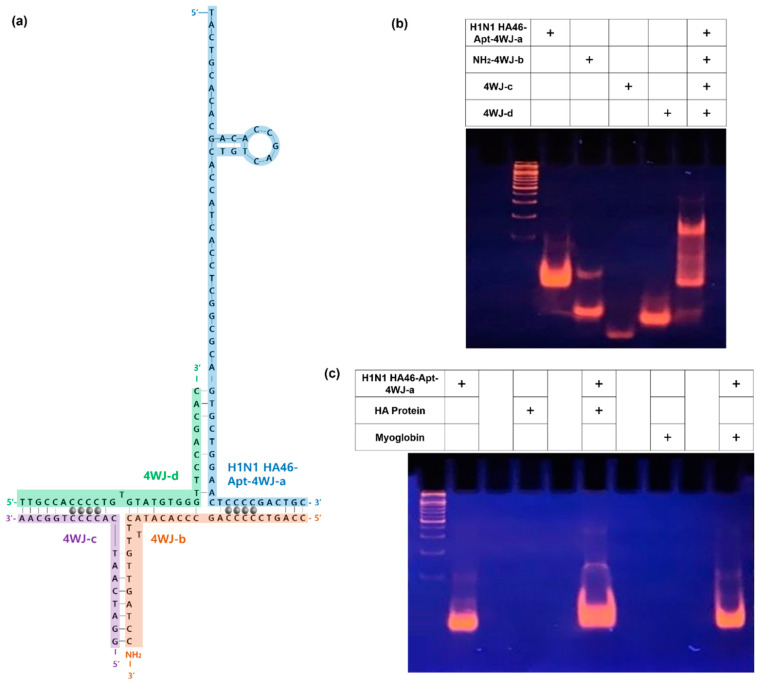
(**a**) Schematic image of expected multifunctional DNA 4WJ structure from 4WJ-a of H1N1 aptamer for detection (blue line), amine-modified 4WJ-b for immobilization (orange line), 4WJ-c for immobilization (purple line), 4WJ-d for immobilization (green line), and intercalated silver ions (silver circles). (**b**) TBM-PAGE gel image 4WJ structure. (**c**) TBE-PAGE gel image of 4WJ-a, HA protein, myoglobin. 4WJ, four-way junction; PAGE, polyacrylamide gel electrophoresis.

**Figure 3 materials-14-00343-f003:**
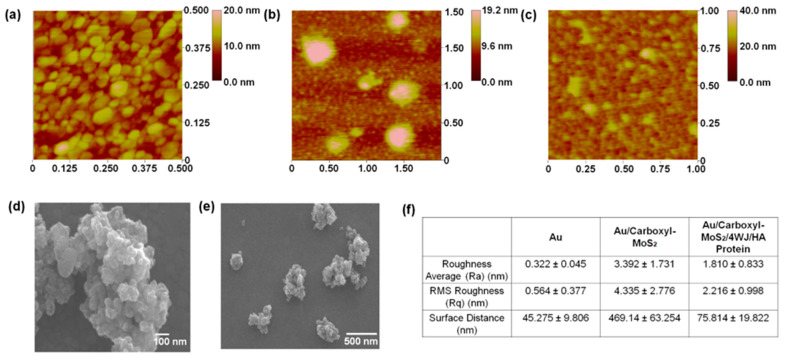
(**a**) surface morphology of bare Au substrate by AFM, (**b**) Surface morphology of carboxyl-MoS_2_ self-assembled Au electrode by AFM, (**c**) Surface morphology of HA protein/DNA 4WJ on carboxyl-MoS_2_ self-assembled Au electrode by AFM. (**d**) Particle analysis of carboxyl-MoS_2_ by FE-SEM, (100 nm scale), (**e**) Particle analysis of carboxyl-MoS_2_ by FE-SEM, (500 nm scale) (**f**) Surface roughness and surface distance analysis of fabricated biofilm.

**Figure 4 materials-14-00343-f004:**
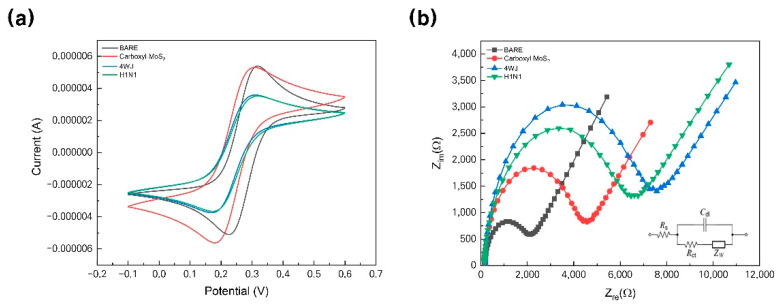
(**a**) Cyclic voltammogram of Au (black line), Au/carboxyl-MoS_2_ (red line), Au/carboxyl-MoS_2_/4WJ (blue line), and Au/carboxyl-MoS_2_/4WJ/HA protein (green line) in 5 mM [Fe(CN)_6_]^3−/4−^ solution in 10 mM HEPES (pH 7.04) containing 1 M KCl at a scan rate of 50 mV/s. (**b**) Impedance spectra of Au (black line), Au/carboxyl-MoS_2_ (red line), Au/carboxyl-MoS_2_/4WJ (blue line), and Au/carboxyl-MoS_2_/4WJ/HA protein (green line) in 5 mM [Fe(CN)_6_]^3−/4−^ solution in 10 mM HEPES (pH 7.04) containing 1 M KCl; the frequency range was from 100 kHz to 1 Hz at a DC potential of 0.25 V. 4WJ, four-way junction.

**Figure 5 materials-14-00343-f005:**
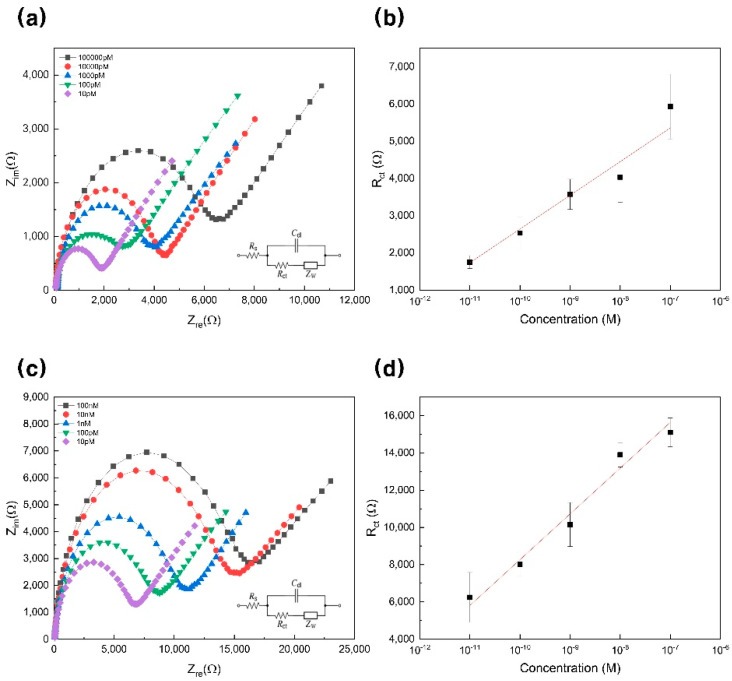
(**a**) Impedance spectra for different concentrations of H1N1 (artificial). (**b**) Linear regression curve of different H1N1 concentrations; linear range from 100 nM to 10 pM. (**c**) Impedance spectra for different concentrations of H1N1 diluted with 10% human serum in 10 mM HEPES and 5 mM [Fe(CN)_6_]^3-/4-^. (**d**) Linear regression curve of different H1N1 concentrations in 10% human serum; linear range from 100 nM to 10 pM. HEPES, 4-(2-hydroxyethyl)-1-piperazineethanesulfonic acid.

**Figure 6 materials-14-00343-f006:**
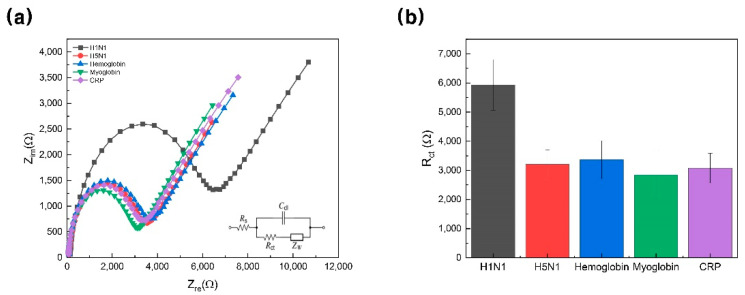
(**a**) Impedance spectra of proposed biosensor with various targets (100 nM), including H1N1 (black line), H5N1 (red line), hemoglobin (blue line), myoglobin (green line), and CRP (purple line) in 5 mM [Fe(CN)_6_]^3−/4−^ solution in 10 mM HEPES (pH 7.04) containing 1 M KCl. (**b**) Change in charge-transfer resistance based on selectivity test (error bars: standard deviation). CRP, C-reactive protein; HEPES, 4-(2-hydroxyethyl)-1-piperazineethanesulfonic acid.

**Table 1 materials-14-00343-t001:** DNA sequences.

DNA	DNA Sequence
H1N1 HA aptamer-tagged 4WJ-a	5′-TAC TGC ACA CGA CAC CGA CTG TCA CCA TCA CCT CGG CGC AGT GCT GGA ACT CCC CGA CTG C-3′
Amine group-tagged 4WJ-b	5′-CCA GTC CCC CAG CCC ACA TAC TTT GTT GAT CC-3′-NH_2_
4WJ-c	5′-GGA TCA ATC ACC CCT GGC AA-3′
4WJ-d	5′-TTG CCA CCC CTG TGT ATG TGG GTT CCA GCA C-3′

**Table 2 materials-14-00343-t002:** Comparison of this study with other studies for H1N1 detection.

Probe	Detection Method	LOD	Linear Range	Ref.
Antibody	LSP	13.9 pg/mL	5 ~ 50 ng/mL	[34]
Antibody	SPR	30 PFU/mL	Not in the linear range	[35]
DNA Aptamer	Fluorescence	3.45 nM	10 ~ 100 nM	[36]
DNA Aptamer	CV/EIS	3.7 PFU/mL	10 ~ 10,000 PFU/mL	[23]
DNA Aptamer	Fluorescence	138 pg/mL	200 pg/mL ~ 200 μg/mL	[37]
Antibody/Antigen	Absorbance	10.79 pg/mL	10 pg/mL ~ 10 μg/mL	[38]
Antibody	CV/EIS	0.5 PFU/mL	1 ~ 10,000 PFU/mL	[39]
DNA Aptamer	CV/EIS	10 pM	10 pM ~ 100 nM	Present Study

## Data Availability

Data is contained within the article.

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
