# Peer review of "Fabrication of Electrochemical Influenza Virus (H1N1) Biosensor Composed of Multifunctional DNA Four-Way Junction and Molybdenum Disulfide Hybrid Material"

_materials, 2021, doi:10.3390/ma14020343_

Round 1

Reviewer 1 Report

The manuscript creates a biosensor composed of multifunctional DNA 4-way junction/molybdenum disulfide hybrid material for the precise virus (H1N1) detection. The author presented a precise method to detect H1N1 with LOD of 100 nM to 10 pM. However, I would suggest the authors revise the manuscript carefully. Some comments are as follows.

  1. The English is really poor. Too many simple grammatical errors and confusing sentences. Proof-read carefully!
  2. Lines 15 and 16 of the article highlight the outbreak of COVID-19 and the importance of specificity testing, but the article does not explain the specificity testing for COVID-19 and H1N1.
  3. There is no fundamental explanation for carboxyl-MoS2 enhanced sensitivity in line 23 of the article.
  4. “Similar results”, “high specificity and selectivity” in line 32, 33 require scientific analysis, confusion matrix, or other scientific analysis.
  5. In line 55 of the introduction, the authors describe a portable H1N1 sensor, but the full text doesn’t describe the portability of the device.
  6. The authors have repeatedly emphasized the high H1N1 specificity of this work, however, electrochemical methods and integration of molybdenum carboxyl disulfide are not strongly correlated with improved specificity, the introduction of silver ions to improve the specificity of the assay requires adequate scientific data to characterize it. The core innovation of this paper needs to be re-emphasized.
  7. The detailed procedures for the extraction and pretreatment of H1N1 samples are missing and need to be added to the Materials and Methods.
  8. MoS2 distribution needs further characterization in Figure 3.
  9. The difference in detection between the artificial sample and the actual sample in Figure 5 needs to be explained fundamentally in the paper.
  10. The scales of the coordinate axes need to be uniform in Figure 5.c, d.

Author Response

REVIEWER REPORT(S):

Reviewer 1.

Comments to the Author

This paper developed the electrochemical H1N1 biosensor comprised of the multi-functional DNA aptamer and Molybdenum Disulfide Hybrid Material. Thank you for considering my article for publication in Materials. I am grateful to you and the reviewers for the valuable suggestions provided.

Here are responses to the reviewer comments:

Comment 1: The English is really poor. Too many simple grammatical errors and confusing sentences. Proof-read carefully!

→ In agreement with the reviewer's opinion, the author proceeded to proofread in English.

Comment 2: Lines 15 and 16 of the article highlight the outbreak of COVID-19 and the importance of specificity testing, but the article does not explain the specificity testing for COVID-19 and H1N1

→ In agreement with the reviewer's opinion, the authors removed the COVID-19 mention from the abstract.

Comment 3: There is no fundamental explanation for carboxyl-MoS2 enhanced sensitivity in line 23 of the article.

→ In agreement with the reviewer's opinion, the author additionally described the content of the sensing system using MoS2 on page 8.

Comment 6: The authors have repeatedly emphasized the high H1N1 specificity of this work, however, electrochemical methods and integration of molybdenum carboxyl disulfide are not strongly correlated with improved specificity, the introduction of silver ions to improve the specificity of the assay requires adequate scientific data to characterize it. The core innovation of this paper needs to be re-emphasized.

→ The authors agree with the reviewer's opinion, based on reviewer’s suggestion, the present study introduced the multi-functional DNA 4WJ intercalated with silver ions on carboxyl-MoS2-modified electrode. So, we emphasized the point the use of nanobio hybrid material can be used to H1N1 virus biosensor application in abstract and conclusion session.

Comment 4: “Similar results”, “high specificity and selectivity” in line 32, 33 require scientific analysis, confusion matrix, or other scientific analysis.

→ In agreement with the reviewers' comments, authors rewrote the line 32, 33 more clearly.

Comment 5: In line 55 of the introduction, the authors describe a portable H1N1 sensor, but the full text doesn’t describe the portability of the device.

→ In agreement with the reviewer's opinion, the author has removed the reference to portable.

Comment 7: The detailed procedures for the extraction and pretreatment of H1N1 samples are missing and need to be added to the Materials and Methods.

→ In this study, authors didn’t use the H1N1 virus sample because of biosafety issue in our country (COVID-19 issue). Authors only used the HA protein which originated from H1N1 coat protein (The influenza A H1N1 (A/New Caledonia/20/1999) Hemagglutinin/HA Protein (His Tag), Sino biology, China). We already described this material in the Materials and Methods.

Comment 8: MoS2 distribution needs further characterization in Figure 3.

→ In agreement with the reviewers, the authors added relevant data to figure 3. We carried out additional FE-SEM image in Figure. 3. Please check the revised manuscript.

Comment 9: The difference in detection between the artificial sample and the actual sample in Figure 5 needs to be explained fundamentally in the paper.

→ In agreement with the reviewer's opinion, the author described the difference between "artificial sample" and "actual sample" in Figure 5.

Comment 10: The scales of the coordinate axes need to be uniform in Figure 5.c, d.

→ In agreement with the reviewer's opinion, the author revised the format in Figure 5.

Reviewer 2.

Comments to the Author

This paper developed the electrochemical H1N1 biosensor comprised of the multi-functional DNA aptamer and Molybdenum Disulfide Hybrid Material. Thank you for considering my article for publication in Materials. I am grateful to you and the reviewers for the valuable suggestions provided.

Here are responses to the reviewer comments:

Comment 1: Materials Science and Engineering: C, Volume 99, June 2019, Pages 511-519, Fabrication of electrochemical biosensor consisted of multi functional DNA structure/porous au nanoparticle for avian influenza virus (H5N1) in chicken serum, TaekLeeaSun YongParkaHongjeJangbGa-HyeonKimaYeonjuLeeaChulhwanParkaMohsenMohammadniaeicMin-HoLeecJunhongMinc

Authors are using their own data, figures in this MS. I suspect Self-plagiarism. Double check Please.

→ Authors disagree with the reviewer's opinion, our previous study (Lee et al., Materials Science and Engineering: C, Volume 99, June 2019, Pages 511-519) was designed to detect the Avian influenza virus (H5N1). To achieve this goal, we introduced DNA 3 way junction which contained H5N1 aptamer, DNAzyme and thiol group. Also, for the signal enhancement, the porous Au nanoparticle was introduced. And the electrode design was totally different compared to present work.

The present work firstly introduced the DNA 4WJ with silver ion intercalated at each two arms. For the target recognition H1N1 aptamer was embodied into the head group of DNA 4WJ and the amine group was introduced for effective immobilization without additional linker. Also, the carboxyl-MoS2 is firstly introduced this work for electrochemical signal facilitation. That is the differences between the present work and previous work.

Comment 2: Schematic in figure 1 is wrong. Working sensor designis confusing and authors highlighted reference electrode. Please change it.

→ In agreement with the reviewer's opinion, the author revised the schematic diagram of the sensor.

Comment 3: Author need to do critical comparison with the performance of known sensors/probes for the analysis of the target analyte species.

→ In agreement with the reviewer's opinion, the authors have added a table for comparison with other H1N1 detection studies (Table 1).

Comment 4: Though an investigation of the sensor response to potentially interfering species has been carried out, the study as well as sensor is not reliable as the performance of the sensor has been directly affected by 50% with high standard deviation. Please see the Figure 6.

→ In agreement with the reviewer's opinion, the author modified and modified the statistical processing method of the data in Figure 6.

Comment 5: Insert a Table in MS with merits of the sensor and discuss it in the main text.

→ In agreement with the reviewer's opinion, the authors have added a table for comparison with other H1N1 detection studies (Table 1) and the advantages of this sensor are described in the conclusion section.

Comment 6: Please prove synergistic effect the MoS2 and its electro catalytic effect on detection system by experimental data as proof of concept.

→ In agreement with the reviewer's opinion, the author additionally described the content of the sensing system using MoS2 on page 8.

Comment 7: Figure 3, AFM images are of not good quality. Author must submit the visibly clear and neat images to journal. This manuscript doesn’t reflect the quality and scientific importance at present condition

→ In agreement with the reviewer's opinion, the author revised the AFM image in Figure 3.

Comment 8: All figures need scientifically approved X and Y labels. Please revise them all.

→ In agreement with the reviewer's opinion, the author revised all figures.

Reviewer 2 Report

MS lacks novelty as authors already reported art of this work by authors in

  1. Materials Science and Engineering: C, Volume 99, June 2019, Pages 511-519, Fabrication of electrochemical biosensor consisted of multi functional DNA structure/porous au nanoparticle for avian influenza virus (H5N1) in chicken serum, TaekLeeaSun YongParkaHongjeJangbGa-HyeonKimaYeonjuLeeaChulhwanParkaMohsenMohammadniaeicMin-HoLeecJunhongMinc
  2. Authors are using their own data, figures in this MS. I suspect Self-plagiarism. Double check Please.
  3. Schematic in figur 1 is wrong. Working sensor designis confusing and authors highlighted reference electrode. Please change it.

    - Author need to do critical comparison with the performance of known sensors/probes for the analysis of the target analyte species.

    -Though an investigation of the sensor response to potentially interfering species has been carried out, the study as well as sensor is not reliable as the performance of the sensor has been directly affected by 50% with high standard deviation. Please see the Figure 6.

    -Insert a Table in MS with merits of the sensor and discuss it in the main text.

    - Please prove synergistic effect the MoS2 and its electro catalytic effect on detection system by experimental data as proof of concept.

    Figure 3, AFM images are of not good quality. Author must submit the visibly clear and neat images to journal. This manuscript doesn’t reflect the quality and scientific importance at present condition.

    All figures need scientifically approved X and Y labels. Please revise them all.

Author Response

REVIEWER REPORT(S):

Reviewer 2.

Comments to the Author

This paper developed the electrochemical H1N1 biosensor comprised of the multi-functional DNA aptamer and Molybdenum Disulfide Hybrid Material. Thank you for considering my article for publication in Materials. I am grateful to you and the reviewers for the valuable suggestions provided.

Here are responses to the reviewer comments:

Comment 1: Materials Science and Engineering: C, Volume 99, June 2019, Pages 511-519, Fabrication of electrochemical biosensor consisted of multi functional DNA structure/porous au nanoparticle for avian influenza virus (H5N1) in chicken serum, TaekLeeaSun YongParkaHongjeJangbGa-HyeonKimaYeonjuLeeaChulhwanParkaMohsenMohammadniaeicMin-HoLeecJunhongMinc

Authors are using their own data, figures in this MS. I suspect Self-plagiarism. Double check Please.

→ Authors disagree with the reviewer's opinion, our previous study (Lee et al., Materials Science and Engineering: C, Volume 99, June 2019, Pages 511-519) was designed to detect the Avian influenza virus (H5N1). To achieve this goal, we introduced DNA 3 way junction which contained H5N1 aptamer, DNAzyme and thiol group. Also, for the signal enhancement, the porous Au nanoparticle was introduced. And the electrode design was totally different compared to present work.

The present work firstly introduced the DNA 4WJ with silver ion intercalated at each two arms. For the target recognition H1N1 aptamer was embodied into the head group of DNA 4WJ and the amine group was introduced for effective immobilization without additional linker. Also, the carboxyl-MoS2 is firstly introduced this work for electrochemical signal facilitation. That is the differences between the present work and previous work.

Comment 2: Schematic in figure 1 is wrong. Working sensor designis confusing and authors highlighted reference electrode. Please change it.

→ In agreement with the reviewer's opinion, the author revised the schematic diagram of the sensor.

Comment 3: Author need to do critical comparison with the performance of known sensors/probes for the analysis of the target analyte species.

→ In agreement with the reviewer's opinion, the authors have added a table for comparison with other H1N1 detection studies (Table 1).

Comment 4: Though an investigation of the sensor response to potentially interfering species has been carried out, the study as well as sensor is not reliable as the performance of the sensor has been directly affected by 50% with high standard deviation. Please see the Figure 6.

→ In agreement with the reviewer's opinion, the author modified and modified the statistical processing method of the data in Figure 6.

Comment 5: Insert a Table in MS with merits of the sensor and discuss it in the main text.

→ In agreement with the reviewer's opinion, the authors have added a table for comparison with other H1N1 detection studies (Table 1) and the advantages of this sensor are described in the conclusion section.

Comment 6: Please prove synergistic effect the MoS2 and its electro catalytic effect on detection system by experimental data as proof of concept.

→ In agreement with the reviewer's opinion, the author additionally described the content of the sensing system using MoS2 on page 8.

Comment 7: Figure 3, AFM images are of not good quality. Author must submit the visibly clear and neat images to journal. This manuscript doesn’t reflect the quality and scientific importance at present condition

→ In agreement with the reviewer's opinion, the author revised the AFM image in Figure 3.

Comment 8: All figures need scientifically approved X and Y labels. Please revise them all.

→ In agreement with the reviewer's opinion, the author revised all figures.

Round 2

Reviewer 1 Report

The manuscript creates a biosensor composed of multifunctional DNA 4-way junction/molybdenum disulfide hybrid material for the precise virus (H1N1) detection. The authors have refined most of the recommendations. The revised version meets the requirements. I am glad to recommend publication.